# Lockdown, relaxation, and acme period in COVID-19: A study of disease dynamics in Hermosillo, Sonora, Mexico

**Mayra R. Tocto-Erazo**[1], **Jorge A. Espíndola-Zepeda**[1], **José A. Montoya-Laos**[1], **Manuel A. Acuña-Zegarra**[1], **Daniel Olmos-Liceaga**[1], **Pablo A. Reyes-Castro**[2], **Gudelia Figueroa-Preciado**[1]*

**1** Departamento de Matemáticas, Universidad de Sonora, Hermosillo, Sonora, México, **2** Centro de Estudios en Salud y Sociedad, El Colegio de Sonora, Hermosillo, Sonora, México

* gudelia.figueroa@unison.mx

**Data Availability Statement:** Data are available in a public repository. According to the Official Diary published by the Mexican Federal Government,

## Abstract

Lockdown and social distancing measures have been implemented for many countries to mitigate the impacts of the COVID-19 pandemic and prevent overwhelming of health services. However, success on this strategy depends not only on the timing of its implementation, but also on the relaxation measures adopted within each community. We developed a mathematical model to evaluate the impacts of the lockdown implemented in Hermosillo, Mexico. We compared this intervention with some hypothetical ones, varying the starting date and also the population proportion that is released, breaking the confinement. A Monte Carlo study was performed by considering three scenarios to define our baseline dynamics. Results showed that a hypothetical delay of two weeks, on the lockdown measures, would result in an early acme around May 9 for hospitalization prevalence and an increase on cumulative deaths, 42 times higher by May 31, when compared to baseline. On the other hand, results concerning relaxation dynamics showed that the acme levels depend on the proportion of people who gets back to daily activities as well as the individual behavior with respect to prevention measures. Analysis regarding different relaxing mitigation measures were provided to the Sonoran Health Ministry, as requested. It is important to stress that, according to information provided by health authorities, the acme occurring time was closed to the one given by our model. Hence, we considered that our model resulted useful for the decision-making assessment, and that an extension of it can be used for the study of a potential second wave.

## Introduction

In late December 2019, a novel coronavirus SARS-CoV-2 (severe acute respiratory syndrome coronavirus 2) was first reported in Wuhan, China [1, 2]. Since then, the pandemic of Coronavirus Disease (COVID-19) has spread in 188 countries, with 21,809,170 millions of infections

COVID-19 data are considered open data and updated daily by Dirección General de Epidemiología at the website https://www.gob.mx/salud/documentos/datos-abiertos-152127.

**Funding:** This work was supported by Consejo Nacional de Ciencia y Tecnología, Project 313269 (GFP). This funding source had no role in the definition of the study design, interpretation or publication of the results.

**Competing interests:** The authors have declared that no competing interests exist.

and 772,452 deaths registered worldwide [3]. Mexico reported its first case in late February 2020, and by the middle of August, public health authorities confirmed around 525,733 infections and more than 57,023 deaths [4]. Based on their clinical manifestations, cases have ranged from mild/moderate to severe, and even some in critical conditions. Severity illness and risk of mortality increase by age and also by the presence of some underlying conditions like hypertension, diabetes, cardiovascular, and cerebrovascular disease [5]. COVID-19 most common symptoms are fever, fatigue, dry cough, myalgia, and severe cases frequently include dyspnea and/or hypoxemia [5–7].

SARS-CoV-2, the virus that causes COVID-19, is highly infectious and spreads predominantly from person-to-person. In the absence of a vaccine or an effective treatment, some non-pharmaceutical community strategies like isolation, testing, contact tracing, and physical distancing have been the main interventions adopted by most of the nations to mitigate this pandemic and reduce the velocity of transmission [8, 9]. From the middle of March to May 30th, Mexican Ministry of Health implemented a National Campaign for Healthy Distance (Jornada Nacional de Sana Distancia), a public health intervention based on physical distancing measures, closing schools as well as non-essential workplaces, and asking for citizens to stay-at-home [10]. However, federal measures demand not only a strong inter-jurisdictional coordination between national, state, and local government levels [11], but also a comprehensive understanding of the disease transmission dynamic, to achieve timely interventions within each locality.

The comprehension of this pandemic has grab the interest of many scientific areas, mainly with the aim of providing ideas that could reduce the severity of the disease. In particular, the area of mathematical modeling has drawn the attention during this epidemic mostly due to its usefulness in providing information about the evolution of transmissible diseases. Current work is focused on parameter estimation that serves as a basis for more complex studies [12], the evaluation of non-pharmacological interventions during the epidemic, such as social distancing or lockdown [13–18] and forecast short term trends of the disease [19]. In general, one of the main purposes of mathematical models has been the evaluation of the effects of different governmental interventions and also providing to decision-makers with more elements for responding to a need, in a more conscious manner [20].

This work aims to evaluate the lockdown and relaxation measures implemented in Hermosillo, Sonora, Mexico. In order to reach our purpose, we developed a mathematical model of the Kermack-McKendrick type, which have been widely used to study COVID-19 disease (e.g. [13–15, 21–23]), using different statistical techniques to estimate some parameter values (e.g. [12–14, 19]). We used some statistical techniques to have the profile of a baseline scenario for being compared with some hypothetical ones, varying the starting date and the population proportion released, breaking the confinement. It is known that COVID-19 predictions are not an easy task, even if data is available from the beginning of the epidemic [24]. Nevertheless, in our case, data availability made possible to uncover robust information that was useful for decision-making.

Our manuscript is organized as follows. Initially, we present our proposed mathematical model. Then, statistical analyses of different parameter scenarios, that validate the data, are presented. A discussion about the results obtained with the adjusted models is included. Our results arise from the statistical and modeling perspectives and are related to the occurring time for the incidence peak of the disease (acme), implications of lockdown occurrence time, and consequences of the lifting mitigation measures. Finally, we end up with a discussion section.

## Methods

### Compartmental mathematical model

We formulate a compartmental mathematical model, whose diagram can be observed in Fig 1, where susceptible ($S$), exposed ($E$), asymptomatic infectious ($I_A$), symptomatic infectious ($I_S$), recovered ($R$), quarantine ($Q$), hospitalized ($H$), and dead individuals ($D$) are considered. $P$ represents a proportion of individuals in the population that decided to stay at home in order to protect themselves from illness, and $P_R$ are those released from the $P$ class, when certain proportion of protected individuals needed or decided to break control measures.

To formulate the mathematical model, we considered that susceptible individuals are moved to the protected class when they obey the mitigation measures implemented by the government and some become infected but not yet infectious (exposed class) when interacting with an infectious individual. Dynamics of protected individuals is similar; that is, they either can become infected when interacting with an infectious individual or moved to the protected released class. This last is a result of a mitigation measures break up (a proportion of the protected population returns to their usual activities). On the other hand, protected released people only leave the class by the interplay with symptomatic or asymptomatic individuals (becoming infected but not yet infectious). The exposed class represents individuals that are infected but not infectious. After a while, an exposed individual can become infectious, asymptomatic, mildly symptomatic, or severe symptomatic. As a first approximation and to analyze data of a specific Mexican state, we considered that the stages previously mentioned are grouped into two classes: i) asymptomatic people ($I_A$), and severe symptomatic people ($I_S$). We assumed that mildly symptomatic people can be distributed in both classes. People from $I_A$ class are recovered with a mean time equal to $1/\eta_a$. In contrast, individuals from $I_S$ class are identified as infected after $1/\gamma_s$ days (on average), after which they are reported and become hospitalized or quarantine/ambulatory. We considered that ambulatory individuals might recover or worsen their condition, being then hospitalized. This happens after $1/\psi$ days (on

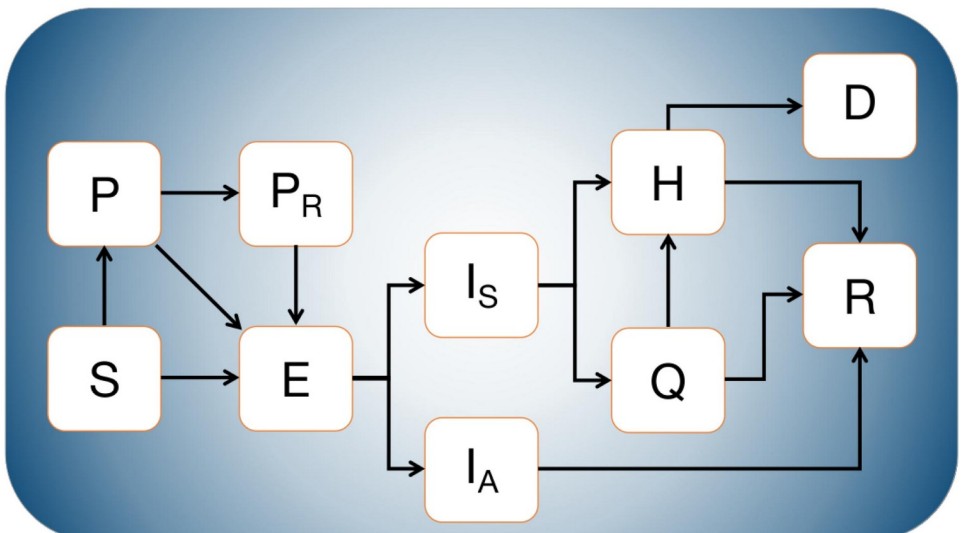

**Fig 1. Flow diagram of the mathematical model.** $S$, $E$, $I_A$, $I_S$, $H$, $Q$, $R$, $D$ represent the populations of susceptible, exposed, asymptomatically infected, symptomatically infected, hospitalized, quarantined, recovered and dead individuals, respectively. Protected individuals ($P$) get involved in the disease dynamics when mitigation measures are implemented, whereas released population ($P_R$) does so when relaxation of these measures occurs.

average). Finally, we assumed that only hospitalized individuals may die, and that occurs after $1/\mu$ days, on average.

Following the hypotheses previously stated, the mathematical model is given by

$$
\begin{aligned}
\dot{S} &= -\left(\frac{\alpha_a I_A + \alpha_s I_S}{N^*}\right)S - \omega_1(t)S \\
\dot{P} &= \omega_1(t)S - \left(\frac{\tilde{\alpha}_a I_A + \tilde{\alpha}_s I_S}{N^*}\right)P - \omega_2(t)P \\
\dot{P}_R &= \omega_2(t)P - \left(\frac{\hat{\alpha}_a I_A + \hat{\alpha}_s I_S}{N^*}\right)P_R \\
\dot{E} &= \left(\frac{\alpha_a I_A + \alpha_s I_S}{N^*}\right)S + \left(\frac{\tilde{\alpha}_a I_A + \tilde{\alpha}_s I_S}{N^*}\right)P + \left(\frac{\hat{\alpha}_a I_A + \hat{\alpha}_s I_S}{N^*}\right)P_R - \delta E \\
\dot{I}_A &= (1-\theta)\delta E - \eta_a I_A \\
\dot{I}_S &= \theta\delta E - \gamma_s I_S \\
\dot{H} &= \beta\gamma_s I_S + \tau\psi Q - \mu H \\
\dot{Q} &= (1-\beta)\gamma_s I_S - \psi Q \\
\dot{R} &= \eta_a I_A + (1-\nu)\mu H + (1-\tau)\psi Q \\
\dot{D} &= \nu\mu H
\end{aligned}
\tag{1}
$$

where $N^* = S + E + I_A + I_S + R + P + P_R$. It is important to emphasize that the infection contact rates of released protected people are less or equal than the infection contact rates of susceptible individuals. On the other hand, $\nu$ and $(1-\nu)$ represent the proportion of hospitalized individuals that recover or die, respectively. Likewise, $\tau$ and $(1-\tau)$ are the proportions of ambulatory individuals who are hospitalized and recovered, respectively. Parameters $\omega_1(t)$ and $\omega_2(t)$ are described in the next subsection, while other parameters definition can be seen in Table 1.

**Modeling the effects of intervention measures.** As happened in other countries, Mexico also implemented control measures to fight against COVID-19. These intervention measures are mainly based on social distancing, in order to reduce contact between people. However, not all Mexican States started these control measures at the same time.

The implementation of social distancing resulted in a proportion of the population being protected by staying at home. For that reason, we modeled this event considering that

**Table 1. System 1 parameter definition and their description.**

| Parameter | Definition |
|---|---|
| $\alpha_a, (\tilde{\alpha}_a, \hat{\alpha}_a)$ | Transmission contact rates for susceptible (protected, protected released) class linked to asymptomatic individuals. |
| $\alpha_s, (\tilde{\alpha}_s, \hat{\alpha}_s)$ | Transmission contact rates for susceptible (protected, protected released) class linked to symptomatic individuals. |
| $\delta$ | Incubation rate. |
| $\theta$ | Proportion of symptomatic individuals. |
| $\eta_a$ | Recovery rate for asymptomatic individuals. |
| $\gamma_s$ | Output rate from the symptomatic class by register. |
| $\beta$ | Proportion of hospitalized individuals. |
| $\psi$ | Output rate from the quarantined class by hospitalization/recovery. |
| $\mu$ | Output rate from the hospitalized class by recovery/death. |

susceptible individuals moved to the protected class during some period. This phenomenon occurs until a certain percentage of the population is reached. We represent this period by $[T_{L_1}, \; T_{U_1}]$. The mathematical description of the dynamics is given by

$$\omega_1(t) = \begin{cases} 0 & , & 0 \leq t < T_{L_1}, \\ w_{10} & , & T_{L_1} \leq t < T_{U_1}, \\ 0 & , & T_{U_1} \leq t, \end{cases} \tag{2}$$

and parameter $w_{10}$ represents the protection rate of susceptible individuals per unit of time. On the other hand, at the moment of writing this paper, it has been observed that many people who were initially obeying mitigation measures have now broken the confinement, going back to their usual activities. For that reason, we consider that certain proportion of protected people become protected released people. We model this phenomenon in a similar way to the one presented in previous function. Thus

$$\omega_2(t) = \begin{cases} 0 & , & 0 \leq t < T_{L_2}, \\ w_{20} & , & T_{L_2} \leq t < T_{U_2}, \\ 0 & , & T_{U_2} \leq t. \end{cases} \tag{3}$$

Here, period from $T_{L_2}$ to $T_{U_2}$ represents the time in which a percentage of the population that breaks the confinement is reached.

**Remark 1** *In order to choose $w_{10}$, we follow the classical population decay equation*

$$\frac{dS}{dt} = -w_{10}S, \tag{4}$$

*and take the value of $w_{10}$ such that a given proportion of individuals in S leaves its class in a given time interval. In other words, we take the solution of* Eq 4 *such that $S(T_{L_1}) = S_{L_1}$, i.e.*

$$S(t) = S_{L_1} e^{-w_{10}(t - T_{L_1})}$$

*Setting $S(T_{U_1}) = kS_{L_1}$, for $k \in (0, 1]$, implies that*

$$w_{10} = \frac{1}{T_{U_1} - T_{L_1}} \ln\left(\frac{1}{k}\right).$$

*Here, $(1 - k)$ is the population proportion that is protected until time $T_{U_1}$. To calculate the value of $w_{20}$, we proceed in a similar manner.*

## Monte carlo study

We performed a Monte Carlo study where different distributions were considered for the parameters included in the mathematical model presented in System 1. The election of these distributions relied not only on the researcher knowledge but also in an extensive search in related literature. Three different scenarios were considered when fitting some epidemic curves derived from this mathematical model (System 1), to the data observed in Hermosillo, Sonora, considering a constraint on the prevalence of COVID-19. As a result of the analyses performed by these three researchers (Scenario 1, Scenario 2 and Scenario 3), we obtained quantile-based intervals, where model parameters can range. These possible parameter values allowed us to explore not only characteristics of the COVID-19 outbreak in Hermosillo,

Sonora, like the acme value and acme date, but also we were able to explore different intervention schemes such as: changes in the beginning and lifting restriction dates, variation in the population proportions that return to usual activities on June 01, 2020, a date fixed by Federal Government.

The Monte Carlo method that was considered here for exploring epidemic characteristics of the COVID-19 outbreak consists of the following steps.

- **Initial conditions for the model:** According to the Mexican National Population Council (CONAPO), projections for 2020 population in Hermosillo is about 930669 people [25]. Regarding the first COVID-19 case registered in Hermosillo by the Sonoran Health System, it occurred in March 16, 2020, being March 11 the onset symptoms date of the first confirmed infected case [26]. In this way, we considered March 11 as the starting date for simulations, with the following initial conditions: $S(0) = 930668$, $I_S(0) = 1$, and $E(0) = I_A(0) = H(0) = D(0) = Q(0) = R(0) = P(0) = P_L(0) = 0$.

- **On-and-off periods of social distancing:** On March 16, date of the first coronavirus case in Sonora, a mandatory confinement was declared by the state governor. This statewide stay-home directive was intended to avoid the spread of this coronavirus. Nevertheless, even on May 6, 2020, Sonora State government divulged a video message asking citizens for remaining in quarantine and taking social distancing seriously, since a considerable increase in the number of cases were occurring.
  Considering the above information, we assumed that the period from March 16 to April 15 was the first period of social distancing, where a considerable proportion of susceptible population became protected, thus $[T_{L_1}, T_{U_1}] = [5, 35]$. A second period was fixed from April 30 to May 15, that is $[T_{L_2}, T_{U_2}] = [50, 65]$. Note that $\omega_1$ and $\omega_2$, in Eqs 2 and 3, have zero dynamic into the interval $(T_{U1}, T_{L2}) = (35, 50)$, that is, neither movement from $S$ to $P$, nor $P$ to $P_R$ is considered. It is important to point out that our motivation for considering periods instead of specific dates for breaking the confinement, is supported by the occurrence of two important dates in Mexico, children's day (April 30) and mother's day (May 10).

- **Model parameter distributions:** We set three different scenarios where different probability distributions were considered for modeling parameters included in System 1. These parameters, as well as their selected distributions, are shown in Table 2.
  Heuristic analysis was used by three different researchers, in order to propose the scenarios presented in Table 2. The selection of the different probability distributions was, in some cases, based on the researcher´s experience, but also this was often closely tied to the versatility of certain distributions to assume that some values of the parameters can occur with lower, equal, or greater probability density. The strategy to delimit the support of these distributions was, either by considering a wide range for parameter values or by selecting these ranges based on a bibliographic review. The fit of the model solutions, to the initially reported data, was done either manually (visual-fit), through a shiny app created in Rstudio (script available on a Github repository [27]), and also by minimizing the sum of squared errors. The values and ranges of model parameters that were taken as a starting point to specify the support of these distributions, are shown in Table 3. Some of these can be found on COVID-19 literature and some others have been assumed by the authors.
  Distributions for $w_{10}$ and $w_{20}$ parameters, were obtained as follows. In Scenario 3, we considered a $\mathcal{U}(0.7, 0.9)$ distribution for the protected proportion of susceptible individuals, and a $\mathcal{U}(0.1, 0.35)$ distribution for the proportion of people who have broken the confinement. Once we have a sample for each one of these proportions, we applied to these samples the equation presented on Remark 1, considering that protected and released population

**Table 2. Model parameter distributions.**

| Parameter | Scenario 1 | Scenario 2 | Scenario 3 |
|---|---|---|---|
| $\alpha_a$ | $\mathcal{N}_{1.22,1.27}(1.2458, 0.0115)$ | $\mathcal{N}_{1.2440,1.5653}(1.3879, 0.125)$ | $\mathcal{N}_{0,\infty}(1.198, 0.05)$ |
| $\alpha_s$ | $\mathcal{N}_{1.10,1.30}(1.2076, 0.0464)$ | $\mathcal{N}_{0.8759,0.9949}(0.9149, 0.06)$ | $\mathcal{N}_{0,\infty}(0.657, 0.05)$ |
| $\tilde{\alpha}_a$ | $\mathcal{N}_{0,0.04}(0.0074, 0.0114)$ | $\mathcal{N}_{0,0.0150}(0.0064, 0.175)$ | $\mathcal{N}_{0,\infty}(0.02, 0.05)$ |
| $\tilde{\alpha}_s$ | $\mathcal{N}_{1.19,1.21}(1.2010, 0.0042)$ | $\mathcal{N}_{0.2889,0.5433}(0.4194, 0.20)$ | $\mathcal{N}_{0,\infty}(0.02, 0.05)$ |
| $\hat{\alpha}_a$ | $\mathcal{U}(0.05, 0.25)$ | $\mathcal{N}_{0.0009,0.0784}(0.0167, 0.045)$ | $\mathcal{N}_{0,\infty}(0.02, 0.05)$ |
| $\hat{\alpha}_s$ | $\mathcal{N}_{0.46,0.66}(0.5839, 0.0509)$ | $\mathcal{N}_{0.7368,0.9789}(0.8772, 0.175)$ | $\mathcal{N}_{0,\infty}(0.02, 0.05)$ |
| $\delta$ | $\mathcal{N}_{0.25,0.35}(0.2923, 0.0211)$ | $\mathcal{N}_{0.1703,0.2529}(0.1990, 0.040)$ | $\mathcal{IG}(25, 5)$ |
| $\theta$ | $\mathcal{U}(0.01, 0.2)$ | $\mathcal{N}_{0.2253,0.3737}(0.2618, 0.085)$ | $\mathcal{U}(0.17, 0.25)$ |
| $\eta_a$ | $\mathcal{N}_{0.08,0.2}(0.1503, 0.025)$ | $\mathcal{N}_{0.04,0.067}(0.0456, 0.02)$ | $\mathcal{IG}(105, 10)$ |
| $\gamma_s$ | $\mathcal{U}(0.5, 2)$ | $\mathcal{N}_{0.7055,1.9143}(1.2415, 0.4)$ | $\mathcal{IG}(3, 1)$ |
| $\beta$ | $\mathcal{U}(0.14, 0.25)$ | $\mathcal{N}_{0.0727,0.1313}(0.1002, 0.055)$ | $\mathcal{B}(8, 50)$ |
| $\tau$ | $\mathcal{U}(0, 0.2)$ | $\mathcal{N}_{0.0277,0.0997}(0.0589, 0.035)$ | $\mathcal{U}(0.1, 0.3)$ |
| $\psi$ | $\mathcal{U}(0.01, 1)$ | $\mathcal{N}_{0.05,0.16}(0.1062, 0.1)$ | $\mathcal{U}(0.06, 0.1)$ |
| $\mu$ | $\mathcal{U}(0.1, 0.8)$ | $\mathcal{N}_{0.0615,0.14}(0.1146, 0.05)$ | $\mathcal{U}(0.05, 0.1)$ |
| $\nu$ | $\mathcal{U}(0.14, 0.3)$ | $\mathcal{N}_{0.0901,0.2472}(0.1550, 0.04)$ | $\mathcal{U}(0.2, 0.4)$ |
| $w_{10}$ | $\mathcal{U}(0.04, 0.08)$ | $\mathcal{N}_{0.0754,0.0864}(0.0809, 0.0075)$ | $\mathcal{U}(0.04, 0.08)$ |
| $w_{20}$ | $\mathcal{U}(0.003, 0.015)$ | $\mathcal{N}_{0.0001,0.0320}(0.0019, 0.0175)$ | $\mathcal{U}(0.007, 0.03)$ |

Here, $\mathcal{N}_{a,b}(\mu_0, \sigma_0)$ is the truncated normal distribution with a truncation range $(a, b)$, where $\mu_0$ and $\sigma_0$ are the mean and variance of this distribution. $\mathcal{IG}(\alpha_0, \beta_0)$ is the inverse gamma distribution with shape and scale parameters $\alpha_0, \beta_0$, respectively. $\mathcal{B}(c, d)$ is the Beta distribution with parameters $c$ and $d$. $\mathcal{U}(\text{min,max})$ is a uniform distribution on an interval that goes from min to max.

**Table 3. Initial parameter ranges and values, taken from current literature or assumed (*) by the researcher.**

| Parameter | Scenario 1 | Scenario 2 | Scenario 3 |
|---|---|---|---|
| $\alpha_a$ | 0.0616–1.5879 [28] | 0.0616–1.5879 [28] | 0.5944–1.68 [17] |
| $\alpha_s$ | 0.0616–1.5879 [28] | 0.0616–1.5879 [28] | 0.5944–1.68 [17] |
| $\tilde{\alpha}_a$ | 0–1.5* | 0.0616–1.5879 [28] | 0–0.5* |
| $\tilde{\alpha}_s$ | 0–1.5* | 0.0616–1.5879 [28] | 0–0.5* |
| $\hat{\alpha}_a$ | 0–1.5* | 0.0616–1.5879 [28] | 0–0.5* |
| $\hat{\alpha}_s$ | 0–1.5* | 0.0616–1.5879 [28] | 0–0.5* |
| $1/\delta$ | 5 [15]; 6 [29]; 0.04–25 [28] | 3.78–6.78 [30]; 3.01–4.91 [30] | 2–14 [31] |
| $\theta$ | 0.10–0.95 [32]; 0.80–0.85 [33] | 0.25–0.90 [32] | 0–0.8* |
| $1/\eta_a$ | 7 [32]; 14 [34] | 22.9–28.1 [35]; 14 [34] | 8.2–15.6 [36] |
| $1/\gamma_s$ | 4 [32]; 1-3.2 [37] | 0.8–8.2 [38] | 0.8–8.2 [38] |
| $\beta$ | 0.04375 [32]; 0.075 [39]; 0.002–0.36 [37] | 0.14 [40] | 0.1–0.3* |
| $\tau$ | 0-0.3* | 0.0277–0.0997* | 0–0.5* |
| $1/\psi$ | 7 [32]; 14 [15] | 8.2–15.6 [36] | 8.2–15.6 [36] |
| $1/\mu$ | 13 [32]; 7 [39] | 3–11 [39] | 4.7-10.3*; 11-25 [41] |
| $\nu$ | 0.125 [32]; 0.42 [37] | 3.8–14.6 [42] | 0–0.4* |
| $w_{10}$ | 0.04–0.08* | 0.0296–0.1206* | 0.04–0.08* |
| $w_{20}$ | 0.003–0.015* | 0.0022–0.0590* | 0.007–0.03* |

proportions are achieved within 30 and 15 days, respectively. The values obtained for $w_{10}$ and $w_{20}$ allowed us to propose the corresponding distributions given in Table 2 as well as the parameters ranges shown in Table 3. A similar procedure was carried out to obtain the distributions for parameters $w_{10}$ and $w_{20}$ in Scenario 1, except that a $\mathcal{U}(0.05, 0.2)$ distribution was considered for the proportion of people who have broken the confinement. For Scenario 2, a $\mathcal{B}(0.8, 0.05)$ distribution was used to describe the protected proportion of susceptible individuals and a $\mathcal{B}(0.05, 0.2)$ distribution for the proportion of people who have broken the confinement. Unlike the other two scenarios, time periods to achieve the percentages of protected and released populations in Scenario 2, are given by $\mathcal{U}(21, 32)$ and $\mathcal{U}(7, 16)$ distributions, respectively. A strategy, similar to the one described for Scenarios 1 and 3, was considered to obtain the distributions given in Table 2 and parameters ranges shown in Table 3.

- **Empirical constraint on prevalence:** A study carried out in Spain shed some light about the highest prevalence percentage in that country, with an estimation around 21.6% [43]. Considering this result, we decided to include solutions where the cumulative number of infected people, since the first case until day 200, were at most 21.6% of the total population in Hermosillo.

- **Data:** The dataset used here is the latest public data on COVID-19, available at the official website of the Mexican Federal Government [26], updated at July 19, 2020. Taking into consideration some decisions adopted by the Mexican government, regarding to lifting confinement measures, the study covered a period spanning from March 11 to May 31. COVID-19 positive cases considered in this study included Hermosillo residents who were registered in a medical unit in the Sonora state. Variables under study were daily cases by onset of symptoms, hospitalized and ambulatory cases by date of admission to a health service unit, and also daily deaths.

- **Empirical restriction on epidemic curves:** In order to ensure reasonable solutions, we considered an inclusion criterion that consists on selecting those solutions such that the sum of squared errors about the data were smaller than some specific upper bound. The reason for adopting this criterion was the fact that epidemiological characteristics were not only determined by the selected scenarios but also were linked to the actual behavior of the epidemic in Hermosillo, Sonora. Next, we briefly describe the steps that were performed to get the upper bounds for the sums of squared errors. First, we obtained $m = 1000$ parameter sets from each scenario and then we used them to calculate daily incidence of symptomatic infections, daily incidence of hospitalized cases, daily incidence of ambulatory cases, and daily incidence of deaths. In order to obtain this information we defined the following variables with respect to the model:

$$
\begin{aligned}
DI_S(k) &= \int_{k-1}^{k} \theta \delta E(t) dt, & DH(k) &= \int_{k-1}^{k} \beta \gamma_s I_S(t) dt, \\
DQ(k) &= \int_{k-1}^{k} (1-\beta) \gamma_s I_S(t) dt, & DD(k) &= \int_{k-1}^{k} \nu \mu H(t) dt,
\end{aligned}
\tag{5}
$$

where $DI_S(k)$, $DH(k)$, $DQ(k)$ and $DD(k)$, are the number of symptomatic infected, hospitalized, ambulatory and death cases, respectively, on the $kth$ day. Then, sums of squared errors were calculated for each selected variable (daily observed incidence of: symptomatic infections, hospitalized cases, ambulatory cases, and deaths) about the theoretical counterpart defined in 5. Then, for each scenario and for each variable, the 25th percentile of the sum of

the squared errors is computed, and considered as an upper bound in the next stage to admit solutions for System 1.

- **Statistics for the analysis:** We obtained its baseline dynamic for each scenario, selecting the 5000 solutions from System 1 that satisfy the criteria previously explained. The R script corresponding to all the carried out calculations is provided in a Github repository [44]. Once these solutions are obtained, we calculated their corresponding 2.5th, 50th, and 97.5th percentiles. It is important to stress that each solution was obtained throughout a particular parameter combination used later to explore other dynamics related to the timing of lockdown implementation and relaxation levels.

## Results

In this section, we applied the methodology previously explained to compute three parameter sets that are used to define our baseline scenarios. Then, the strengths and weaknesses of our results are discussed. Finally, we explored some scenarios regarding possible consequences of i) change of dates for implementing mitigation measures, and ii) lifting mitigation measures on June 01, 2020.

### About the acme occurring time

Based on the three scenarios previously considered, we obtained the quantile-based intervals shown in Table 4. Fig 2 increased our knowledge about the parameters behavior, providing, for each parameter and each scenario, some interesting complementary information. In these plots we can observe that, in some cases, there is no intersection between these empirical distributions, while in others a considerable overlap occurs. This illustrates a well known problem of parameter identifiability where basically, different parameter regions can provide solutions

**Table 4. Median and 95% quantile-based intervals for parameters in System 1.**

|  | Scenario 1 | | | Scenario 2 | | | Scenario 3 | | |
|---|---|---|---|---|---|---|---|---|---|
|  | 2.5% | 50% | 97.5% | 2.5% | 50% | 97.5% | 2.5% | 50% | 97.5% |
| $\alpha_a$ | 1.2251 | 1.2458 | 1.2657 | 1.2991 | 1.4573 | 1.5577 | 1.1102 | 1.2041 | 1.3017 |
| $\alpha_s$ | 1.1239 | 1.2067 | 1.2846 | 0.8795 | 0.9305 | 0.9903 | 0.5645 | 0.6601 | 0.7553 |
| $\tilde{\alpha}_a$ | 0.0007 | 0.0114 | 0.0304 | 0.0005 | 0.0082 | 0.0147 | 0.0019 | 0.0342 | 0.1017 |
| $\tilde{\alpha}_s$ | 1.1931 | 1.2010 | 1.2085 | 0.2981 | 0.4270 | 0.5374 | 0.0018 | 0.0415 | 0.1241 |
| $\hat{\alpha}_a$ | 0.0557 | 0.1517 | 0.2445 | 0.0018 | 0.0342 | 0.0752 | 0.0018 | 0.0371 | 0.1129 |
| $\hat{\alpha}_s$ | 0.4876 | 0.5796 | 0.6526 | 0.7445 | 0.8643 | 0.9742 | 0.0019 | 0.0417 | 0.1242 |
| $\delta$ | 0.2573 | 0.2927 | 0.3317 | 0.1789 | 0.2223 | 0.2510 | 0.1604 | 0.2157 | 0.3049 |
| $\theta$ | 0.0194 | 0.1072 | 0.1961 | 0.2274 | 0.2748 | 0.3634 | 0.1720 | 0.2109 | 0.2481 |
| $\eta_a$ | 0.1013 | 0.1470 | 0.1909 | 0.0404 | 0.0479 | 0.0638 | 0.0792 | 0.0954 | 0.1158 |
| $\gamma_s$ | 0.5447 | 1.2334 | 1.9548 | 0.7298 | 1.0890 | 1.7143 | 0.1294 | 0.3300 | 1.0376 |
| $\beta$ | 0.1424 | 0.1877 | 0.2458 | 0.0751 | 0.1049 | 0.1295 | 0.0769 | 0.1386 | 0.2293 |
| $\tau$ | 0.0050 | 0.0885 | 0.1933 | 0.0304 | 0.0636 | 0.0969 | 0.1057 | 0.1972 | 0.2945 |
| $\psi$ | 0.0207 | 0.2466 | 0.4889 | 0.0532 | 0.1065 | 0.1573 | 0.0610 | 0.0797 | 0.0990 |
| $\mu$ | 0.1141 | 0.4374 | 0.7810 | 0.0648 | 0.1067 | 0.1382 | 0.0512 | 0.0744 | 0.0985 |
| $\nu$ | 0.1435 | 0.2067 | 0.2926 | 0.1048 | 0.1632 | 0.2314 | 0.2058 | 0.2944 | 0.3941 |
| $w_{10}$ | 0.0484 | 0.0590 | 0.0738 | 0.0755 | 0.0786 | 0.0854 | 0.0519 | 0.0631 | 0.0776 |
| $w_{20}$ | 0.0033 | 0.0089 | 0.0147 | 0.0007 | 0.0121 | 0.0302 | 0.0076 | 0.0191 | 0.0294 |

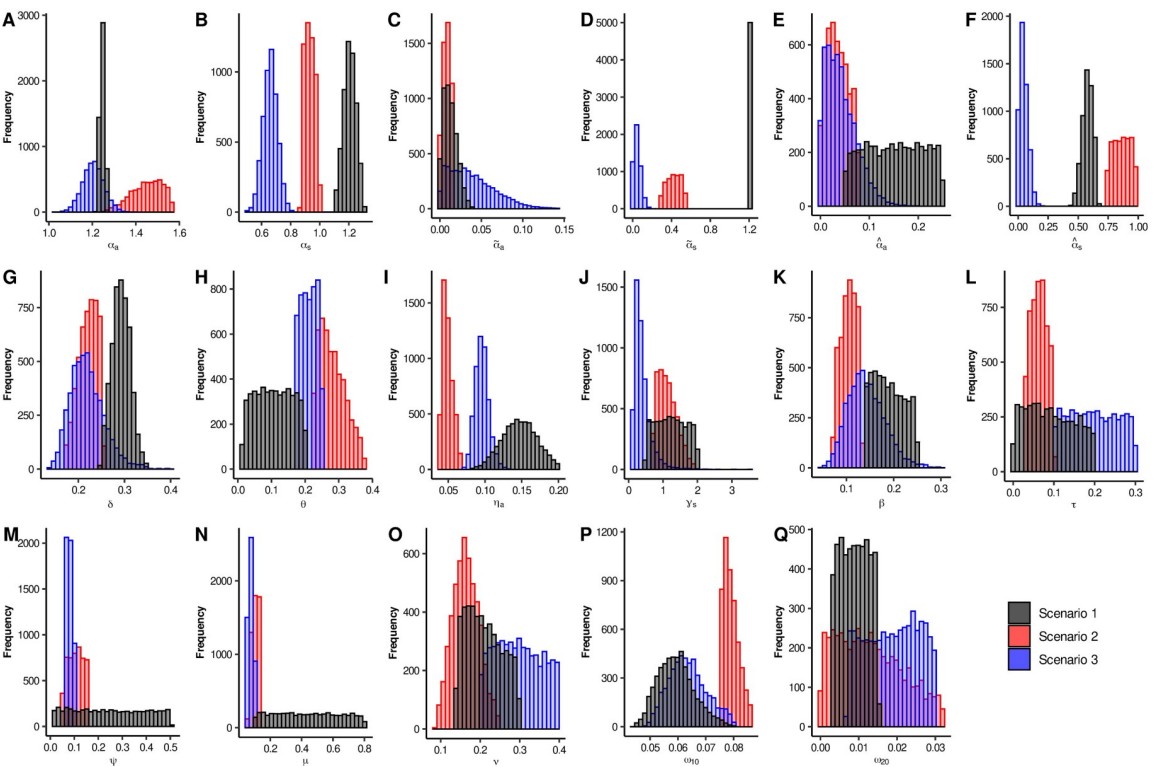

**Fig 2. Histogram for each one of the parameters of System 1.** Black, red and blue bars are related to Scenarios 1, 2, and 3, respectively.

that could give rise the observed data; this can actually be observed in Fig 3. Thus, we must be aware that when models involve too many free parameters, different sets of parameters can also provide a good fit to the data.

To complement the fact previously exposed, we included Fig 4, where the distribution of the maximum number of daily new reported, hospitalized, and deaths can be observed. As expected, the three scenarios provided solutions that in general do not coincide in the acme levels. However, our study gave us some certainty in another aspect. For the three scenarios,

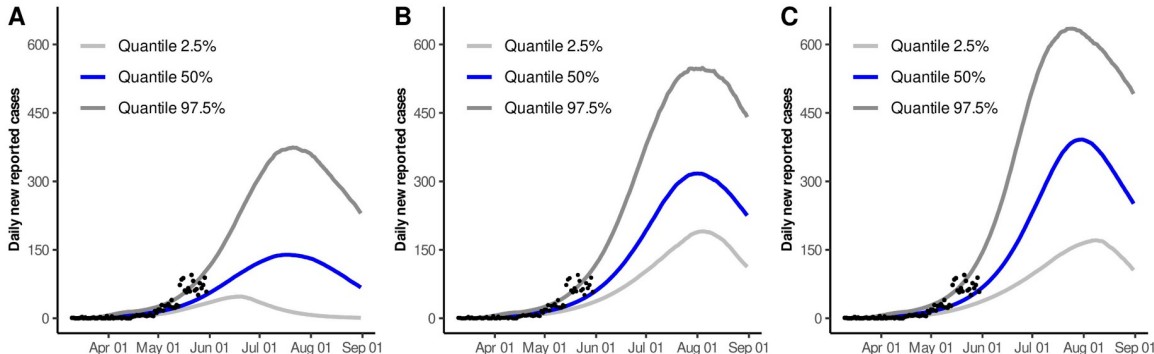

**Fig 3. 95% quantile-based intervals and median estimates for the solution curve of daily new reported cases.** A) Dynamics for Scenario 1. B) Dynamics for Scenario 2. C) Dynamics for Scenario 3. Black dots represent available data from March 11 to May 31 (accessed on July 19).

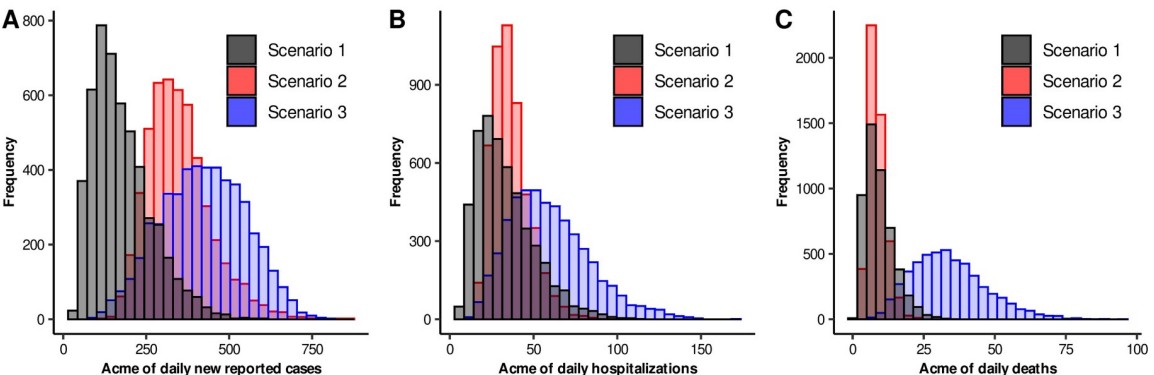

**Fig 4. Histograms of acme levels for different epidemic curves.** Black, red and blue bars are related to Scenarios 1, 2, and 3, respectively. A) acme of daily new reported cases. B) acme of daily hospitalizations. C) acme of daily deaths.

Fig 5 shows the distributions of the estimated date of the acme for the daily new reported, hospitalized, and death variables. Here, we can clearly observe that Scenarios 2 and 3 presented very similar distributions for these three variables. In contrast, histograms for Scenario 1 are flattened, their beginning is too early, and they ended almost at the same dates of Scenarios 2 and 3. Actually, 95% quantile-based intervals for Scenario 1 will almost contain the ones corresponding to Scenarios 2 and 3. These results indicate that even when parameters do not provide consistent information about the intensity of the outbreak, it did preserve the property of having an acme occurring time in a specific time interval.

## Implications of lockdown occurrence time

Based on System 1 and the parameter ranges and values obtained in previous section, we evaluated implications on the magnitude of the variables of interest, if the lockdown had been implemented one or two weeks later than our real scenario. This exploration intends to analyze the possible consequences of a late decision making. For our simulations, we used 5000 parameter combinations of scenario 2, and calculated the quantile 0.5 of all these solutions. Here, we basically present hospitalized prevalence, in order to relate this with bed saturation and cumulative deaths.

It is important to have in mind that in real setting, lockdown took place from March 16 to April 15 (Baseline). Therefore, we carried out these simulations, for Scenario 2, considering

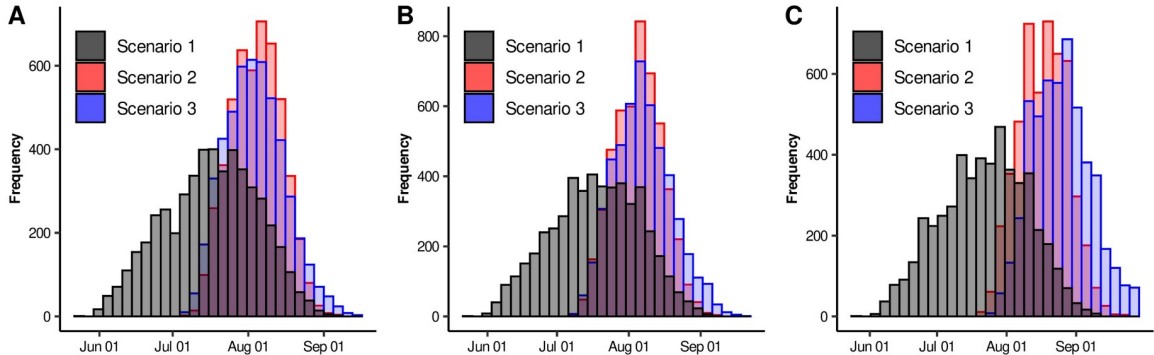

**Fig 5. Histograms of acme dates for different epidemic curves.** Black, red and blue bars are related to Scenarios 1, 2, and 3, respectively. A) acme of daily new reported cases. B) acme of daily hospitalizations. C) acme of daily deaths.

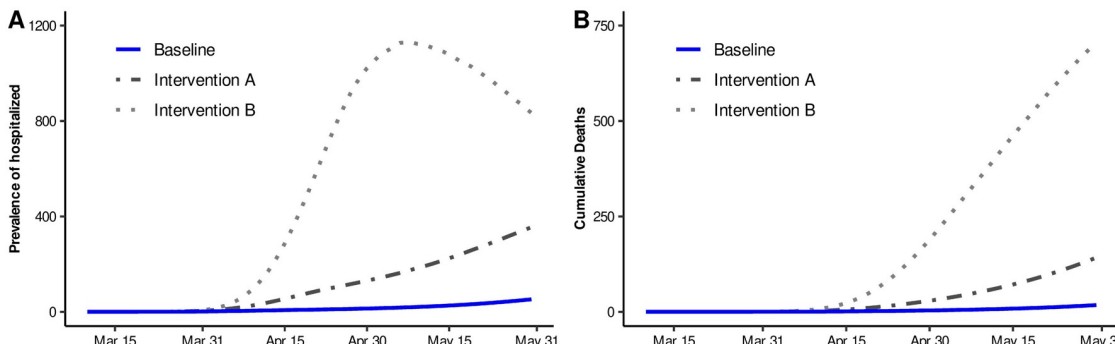

**Fig 6. Median estimates for some epidemic curves.** Blue solid line represents our baseline dynamics for Scenario 2. Dotted-dashed line represents Intervention A, whose mitigation measures are supposed to start on March 23. Dotted line represents Intervention B, whose mitigation measures are assumed to begin on March 30. A) Prevalence of hospitalized. B) Cumulative deaths.

that lockdown took place over a time interval from March 23 to April 22 (Intervention A) and also from March 30 to April 29 (Intervention B). Fig 6 shows the solution for Baseline and these interventions, for Scenario 2. From the figure, we can observe that a considerable increase in the number of daily new hospitalizations and deaths would occur if distancing measures were taken two weeks after the original date, exhibiting the importance of timely decision making. Corresponding results of Scenarios 1 and 3 can be seen in the Section S2 of the S1 File.

## Possible consequences of lifting mitigation measures

On June 01, 2020, Mexican Federal Government established an epidemiological panel. The purpose of this panel was to assign a color (red, orange, yellow, green) to each one of the states of Mexico, and gradually lift mitigation measures, depending on the color assigned to each one of the states. However, as a result of this strategy, an unknown number of people returned to their usual activities since June 01, 2020, independently of the color that this panel assigned to a state. Motivated by this fact, that also occurred in Hermosillo, Sonora, we explored possible consequences that lifting mitigation measures could have on daily new cases, daily new hospitalizations and daily deaths.

Fig 7 shows some epidemic curves under Scenario 2. Here, each curve represents quantile 0.5 of all solutions when considering different proportions of individuals returning to usual activities on June 01, 2020. Baseline curve (solid blue line) represents disease dynamics without lifting mitigation measures. The scenarios named Lifting A, B, and C were constructed considering that approximately 16%, 33%, and 66% of the population, that fulfilled with social-distancing measures, returned to their usual activities on June 01, 2020, respectively. Here, we can deduce that the number of people returning to their usual activities, is directly affecting the acme level of these epidemic curves, which also depend on the adopted social-distancing measures. The latter will be discussed on S1 File. Finally, it is important to have in mind that the Monte Carlo study considered data reported up to May 31, since in June 01, mitigation measures were relaxed, causing an increase in mobility. For scenario 2, when comparing our results with the data reported up to August 14, a poor fitting can be observed in some periods. However, it is important to mention that there is a delay in the information reported by the Federal Government since, as of August 14, this entity reported 8535 cumulative infected, while the Sonora Government reported 11362 cumulative infected. The latter makes us think

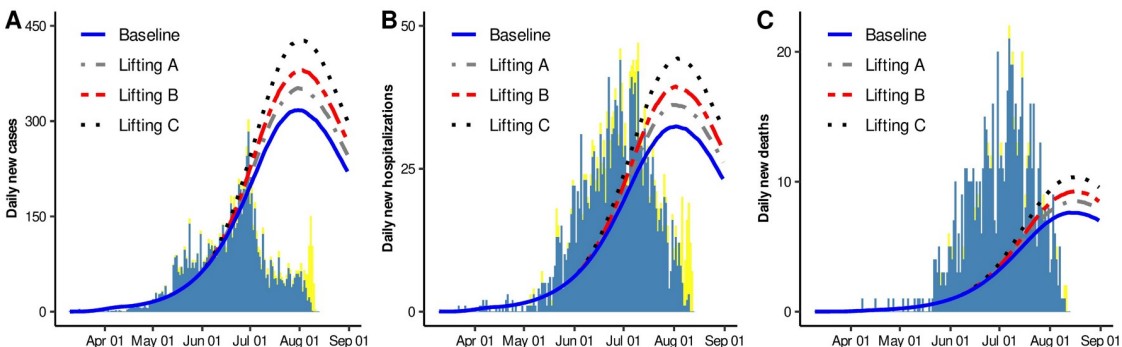

**Fig 7. Median estimates for A) Daily new cases, B) Daily new hospitalizations, and C) Daily new deaths.** Blue solid line represents our baseline dynamics. Grey dotted-dashed, red dashed and black dotted lines represent that approximately 16%, 33%, and 66% of the population that fulfilled with social-distancing measures returned to their usual activities on June 01, 2020, respectively. Blue and yellow bars represent confirmed and suspected+confirmed data for Hermosillo, Sonora, Mexico (data accessed on August 12).

that the adjustment presented for the new daily cases is good. Section S3 of the S1 File. includes these analyses under Scenarios 1 and 3, and similar characteristics were observed.

## Discussion

The comprehension of the COVID-19 epidemic has become a major interest area of study due not only to the lives that have been lost worldwide but also to the economic damages that it caused in different regions in the world. Nowadays, data availability about this epidemic has allowed to show how different mathematical models and statistical techniques are useful for providing valuable information related to decision making, in many particular regions. Moreover, these models are setting the basis for preventing and controlling more catastrophic scenarios in a possible second wave or under the presence of a different propagating virus.

In particular, being researchers in a university where social responsibility is a core mission, our interest was not only focused on the problem of building models to understand the COVID epidemic, but also to provide to the Sonoran Health Ministry with insights regarding the possible consequences of an immediate return to the daily activities, as requested. Then our goal, as in many other places of the world, was to explore strategies for releasing population to their dailyactivities, being always in control about the number of seriously ill individuals and the availability of hospital facilities. Our studies were merely from the epidemiological point of view and the results were valuable for the comprehension of how epidemic curves are affected by changes in confinement, social distance measures and also the proportion of people that can be considered protected [45]. In the current study, we extended our research question when exploring different timing for control measures implementation and proportion of released population. However, the government decision about released individuals actually depended not only on epidemiological factors.

Mathematical models, like the one presented here, are useful to understand some qualitative properties of the evolution of a disease. In that sense, in this work, we proposed a mathematical model to study COVID-19 dynamics in Hermosillo, Sonora, Mexico. Here, we assessed the timing to implement different social-distancing scenarios during COVID-19 epidemic and explored different levels of mitigation-measure relaxation. In order to obtain our baseline dynamics, we conducted a Monte Carlo study, and under three different scenarios, some epidemic curves were fitted. This methodology helped us to set three-parameter distribution sets that adjust the data.

Our results showed that the three scenarios agreed on what is called the acme occurring time, so there might be information contained in the structure of the epidemic and in the model itself that might lead us to observe these results. According to our findings, the median dates for the acme of incidence cases would occur between July 18 and August 6. These results were consistent with ongoing surveillance data provided by local health authorities, which reported the incidence peak by the 31st epidemiological week (July 27—August 2). Since then, decrements have been observed for incidence cases and hospitalizations [46]. In summary, our model described well the epidemic dynamic and the impact of lockdown intervention measures throughout time.

On the other hand, our findings also suggest that a hypothetical delay of two weeks (intervention B) for the implementation of the lockdown measures would result in an early peak (May 9). Moreover, the two weeks delay considered in intervention B, would increment in about 42 times the cumulative deaths, when compared to the ones observed under our baseline, by May 31. In the absence of a vaccine or an effective treatment, the implementation of social distancing measures at the early stages of this pandemic, helped to delay and slowdown the epidemic dynamic, allowing to gain time to strengthen healthcare capacities, avoiding being overwhelmed by an excessive demand.

Regarding lifting mitigation measures it was shown that changes in daily cases, hospitalizations, and deaths, depended on the proportion of people released to the public space on June 01. Fig 7A shows that acme levels varied from 11% to 35% at the peak of the outbreak compared to baseline. Our conclusion regarding this issue must be conservative, since these results clearly depend on the population proportion who returned to usual activities. An important factor that influences on the magnitude of this proportion, is the poverty level [47] and according to official data, 35% of the occupied labor force have informal jobs [48], and 19% lives in poverty condition [49]. Economic inequalities contribute to impeding that a significant proportion of people could maintain a rigorous lockdown, since their conditions force them to return to work. Improvements not just in surveillance but social data, at local level, will benefit future estimations.

It is important to highlight, that some differences concerning the median and quantile-based intervals for the solution curve of daily new reported cases were observed between Scenarios 1, 2 and 3 (see Fig 3 in this document and S1 Fig inside S1 File). Similar behaviors regarding the intensity of the peaks can be observed when comparing Fig 7 with S3 Fig, inside S1 File. In that sense, it is meaningless to talk about predictability on the intensity of the outbreak, when using a model like the one considered here. However, additional specific pieces of information, could be useful to discriminate spurious solutions and head toward having a predictive nature of these results. For example, in Scenario 3, the results claim that released protected, and protected people are infecting practically at the same rate. In other words, people that have returned to their daily activities are protecting themselves as if they still were in the protected class. For this Scenario, results did not show variability in the peaks intensity, when considering different proportions of individuals released on June 01 (see S3 Fig in S1 File).

As a final note, the inclusion of the vital dynamics in the model can be useful when studying the evolution of the disease for longer periods. For example, it can be helpful to provide qualitative information on a possible second outbreak that might occur during the flu season, which runs from September to January.

## Supporting information

**S1 File.**
(PDF)

## Acknowledgments

We thank the anonymous reviewers since their valuable comments and suggestions helped us to improve and clarify this manuscript.

## Author Contributions

**Conceptualization:** José A. Montoya-Laos, Manuel A. Acuña-Zegarra, Daniel Olmos-Liceaga, Pablo A. Reyes-Castro, Gudelia Figueroa-Preciado.

**Data curation:** Mayra R. Tocto-Erazo.

**Formal analysis:** Mayra R. Tocto-Erazo, Jorge A. Espíndola-Zepeda, José A. Montoya-Laos.

**Funding acquisition:** Gudelia Figueroa-Preciado.

**Methodology:** Mayra R. Tocto-Erazo, Jorge A. Espíndola-Zepeda, José A. Montoya-Laos, Manuel A. Acuña-Zegarra, Daniel Olmos-Liceaga.

**Project administration:** José A. Montoya-Laos.

**Software:** Mayra R. Tocto-Erazo, Jorge A. Espíndola-Zepeda, José A. Montoya-Laos, Manuel A. Acuña-Zegarra.

**Supervision:** José A. Montoya-Laos, Manuel A. Acuña-Zegarra.

**Visualization:** Mayra R. Tocto-Erazo, Jorge A. Espíndola-Zepeda.

**Writing – original draft:** Mayra R. Tocto-Erazo, Jorge A. Espíndola-Zepeda, José A. Montoya-Laos, Manuel A. Acuña-Zegarra, Daniel Olmos-Liceaga, Pablo A. Reyes-Castro, Gudelia Figueroa-Preciado.

**Writing – review & editing:** José A. Montoya-Laos, Manuel A. Acuña-Zegarra, Daniel Olmos-Liceaga, Pablo A. Reyes-Castro, Gudelia Figueroa-Preciado.

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
