## [Decision Letter · Decision Letter 0]

16 Oct 2020

PONE-D-20-28007

Lockdown, relaxation, and ACME period in COVID-19: A study of disease dynamics in Hermosillo, Sonora, Mexico

PLOS ONE

Dear Dr. Figueroa-Preciado,

Thank you for submitting your manuscript to PLOS ONE. After careful consideration, we feel that it has merit but does not fully meet PLOS ONE’s publication criteria as it currently stands. Therefore, we invite you to submit a revised version of the manuscript that addresses the points raised during the review process.

We look forward to receiving your revised manuscript.

Kind regards,

Laurent Pujo-Menjouet

Academic Editor

PLOS ONE

Additional Editor Comments:

Dear authors,

following the reviewers suggestions, I am pleased to accept

your manuscript after answering the minor concerns.

Best,

Laurent Pujo-Menjouet

Journal Requirements:

Reviewers' comments:

Reviewer's Responses to Questions

**Comments to the Author**

1. Is the manuscript technically sound, and do the data support the conclusions?

Reviewer #1: Yes

Reviewer #2: Partly

2. Has the statistical analysis been performed appropriately and rigorously? 

Reviewer #1: Yes

Reviewer #2: I Don't Know

3. Have the authors made all data underlying the findings in their manuscript fully available?

Reviewer #1: Yes

Reviewer #2: No

4. Is the manuscript presented in an intelligible fashion and written in standard English?

Reviewer #1: Yes

Reviewer #2: Yes

5. Review Comments to the Author

Reviewer #1: In this paper, the authors studied COVID-19 in Hermosillo, Mexico by using a mathematical model. The model is well-constructed and, in particular, it takes into account the effect of intervention measures by using functions w1 and w2, which represent the transition rates from susceptible to protected and protected to released, respectively. The Monte Carlo study was performed by using parameter sets based on three different scenarios, and estimation results were obtained. Based on these results, the authors discussed the effects of lockdown and lifting mitigation measures. The paper is totally very well-written and I recommend this paper for publication. The mathematical model in this paper would help the readers to evaluate situations that did not happen in reality.

Minor issues

1. Should the authors explain the meaning of the abbreviation ACME?

2. Should the authors estimate the (basic or effective) reproduction numbers?

Reviewer #2: In this article, the authors develop a Kermack-McKendrick-type mathematical model in order to evaluate the confinement and relaxation measures implemented at Hermosillo (Sonora, Mexico). The phenomenon studied is of great interest and topicality, the model chosen is fairly standard, however some points deserve to be explained, such as the choice of different distributions, and also the details of the calculations.

- The introduction requires some references dealing with the mathematical and statistical study of this type of models.

- The model is quite simple, the choice of $\\omega_{0i}$ must be justified, remark 1 is not sufficient.

- The choice of different distributions in a single scenario is to be justified.

- The detailed calculation for each step is necessary to validate the results obtained.

- In the initial conditions of the model, the authors say that the first case of COVID-19 recorded in Hermosillo by the Sonoran Health System, occurred on March 16, 2020 (line 128), and they consider March 11 as the start date of simulations, with the following initial condition: $I_S(0)=1$? Where does this value come from (quote a reference)?

- At line 140, the authors assume that the first period of social distancing is $[T_{L1},T_{U1}]=[5,35]$ and the second period is $[T_{L2},T_{U2}]=[50,65]$, but between $T_{U1}=35$ and $T_{L2}=50$ there is no information, we need details for this period.

- In the section model parameter distributions, the authors established three different scenarios where different probability distributions were taken into account for the modeling parameters included in the mathematical model. How the authors choose the different probability distributions, I would recommend explaining their choice of distributions for the selected parameters in order to understand this part.

- In line 161 (respectively 169), the distributions for the parameters $\\ omega_{10}$ and $\\omega_{20}$ considered in scenario 3 (respectively scenario 1): $U(0.7,0.9)$ and $U(0.1,0.35)$ (respectively $U(0.05,0.2)$), are not the same as those cited in table 2, why? also in the title of table 2, for a normal and uniform distribution, the parameters $a$ and $b$ are used, can you specify if the same parameters are used for both distributions or not.

- The general organization of the paper seems rather chaotic. In section 2 (respectively 3), the authors present the model (respectively some results), but to understand what the diagram compartment is (respectively what is the basis of these results), the reader must go to the end of the article to see the numbers. I highly recommend rearranging the paper in the usual style: each figure should be placed right after its discussion paragraph.

In conclusion, in the given form I cannot recommend the given manuscript for publication. In my opinion, a complete revision of the presentation of the content is needed.

6. PLOS authors have the option to publish the peer review history of their article (what does this mean?). If published, this will include your full peer review and any attached files.

Reviewer #1: No

Reviewer #2: No

---

## [Author Response · Author response to Decision Letter 0]

30 Oct 2020

Reviewer 1: We have incorporated all your constructive comments in the revised manuscript.

Reviewer 2: All your helpful comments and suggestions have been incorporated in the revised manuscript.

---

## [Decision Letter · Decision Letter 1]

13 Nov 2020

Lockdown, relaxation, and acme period in COVID-19: A study of disease dynamics in Hermosillo, Sonora, Mexico

PONE-D-20-28007R1

Dear Dr. Figueroa-Preciado,

We’re pleased to inform you that your manuscript has been judged scientifically suitable for publication and will be formally accepted for publication once it meets all outstanding technical requirements.

Kind regards,

Laurent Pujo-Menjouet

Academic Editor

PLOS ONE

Additional Editor Comments (optional):

Dear colleagues,

following the reviewers decision,

I am pleased to suggest to accept the paper.

Best regards,

Laurent

Reviewers' comments:

Reviewer's Responses to Questions

**Comments to the Author**

1. If the authors have adequately addressed your comments raised in a previous round of review and you feel that this manuscript is now acceptable for publication, you may indicate that here to bypass the “Comments to the Author” section, enter your conflict of interest statement in the “Confidential to Editor” section, and submit your "Accept" recommendation.

Reviewer #2: All comments have been addressed

2. Is the manuscript technically sound, and do the data support the conclusions?

Reviewer #2: Yes

3. Has the statistical analysis been performed appropriately and rigorously? 

Reviewer #2: Yes

4. Have the authors made all data underlying the findings in their manuscript fully available?

Reviewer #2: Yes

5. Is the manuscript presented in an intelligible fashion and written in standard English?

Reviewer #2: Yes

6. Review Comments to the Author

Reviewer #2: In this article, the authors develop a Kermack-McKendrick-type mathematical model in order to evaluate the confinement and relaxation measures implemented at Hermosillo (Sonora, Mexico). The phenomenon studied is of great interest and topicality.

In conclusion, I recommend the given manuscript for publication in journal PLOS ONE.

7. PLOS authors have the option to publish the peer review history of their article (what does this mean?). If published, this will include your full peer review and any attached files.

Reviewer #2: No

---

## [Editor Report · Acceptance letter]

20 Nov 2020

PONE-D-20-28007R1 

Lockdown, relaxation, and acme period in COVID-19: A study of disease dynamics in Hermosillo, Sonora, Mexico  

Dear Dr. Figueroa-Preciado:

I'm pleased to inform you that your manuscript has been deemed suitable for publication in PLOS ONE. Congratulations! Your manuscript is now with our production department. 

Kind regards, 

on behalf of

Dr. Laurent Pujo-Menjouet 

Academic Editor

PLOS ONE